# Dynamic Analysis of a Planar Suspension Mechanism Based on Kinestatic Relations

Guofeng Zhou [1], Shengye Jin [1], Yafei Wang [2,*], Zhisong Zhou [2] and Shouqi Cao [1]

1   College of Engineering Science and Technology, Shanghai Ocean University, Shanghai 201306, China; gfzhou@shou.edu.cn (G.Z.); m210801333@st.shou.edu.cn (S.J.); sqcao@shou.edu.cn (S.C.)
2   School of Mechanical Engineering, Shanghai Jiao Tong University, Shanghai 200240, China; zhouzhisong1993@sjtu.edu.cn
*   Correspondence: wyfjlu@sjtu.edu.cn

**Abstract:** The dynamic characteristics of a vehicle are significantly influenced by the suspension mechanism. In this paper, the nonlinear kinestatic relations of a planar suspension mechanism are taken into account in the dynamic analysis of a vehicle. A planar suspension mechanism can be considered a 1-DOF parallel mechanism. The Jacobian is used for the kinestatic analysis of the suspension. The motions of the suspension can be represented by instantaneous screw. Based on these kinematic and static relations, the dynamic performances of a quarter-vehicle model with a planar suspension mechanism are described in terms of Lagrangian equations. Finally, as illustrated in the examples, two different kinds of road disturbances are inputted into the wheel. The dynamic responses of a quarter-vehicle model are simulated and compared with the simulation software Adams/View for the validity of the theoretical method.

**Keywords:** planar suspension mechanism; quarter-vehicle model; double-wishbone suspension; instantaneous screws; kinestatic analysis; dynamic analysis

## 1. Introduction

Automated driving has emerged as remarkable technological developments. The vehicle dynamic model plays an important role in the longitudinal and lateral motion control for the improvement of the ride comfort and safety of automated driving. The high-confidence vehicle dynamic modelling has always been the challenge of automated driving virtual simulation. Single-track bicycle model and two-track vehicle model are widely used in the longitudinal and lateral control of automated driving. Landolfi et al. [1] proposed an MPC strategy for the connected and automated vehicle using the single-track model. Yu et al. [2] used an extended bicycle model that took the terrain topology into consideration in MPC. Villano et al. [3] adopted a double-track vehicle model for the vehicle sideslip angle estimation. Brinkschulte [4] suggested an efficient nonlinear two-track model for the development of a vehicle simulator. In addition, the roll motion of a vehicle is also one of the most important motions of a vehicle. Li et al. [5] determined an ideal torsion bar coupling the front and rear sprung based on the concept of roll center. Belrzaeg et al. [6] introduced the vehicle planar models and full-vehicle models which have been widely adopted for vehicle motion control. Ataei et al. [7] developed a novel general reconfigurable vehicle dynamic model coupling the longitudinal, lateral and roll motions. However, these vehicle models neglected the effects of the suspension mechanism on the dynamic performances of a vehicle.

Much research has studied the kinematic and static relations of a suspension mechanism towards a significant improvement of a vehicle. Simionescu and Beale [8] presented an optimum method of a multi-link suspension mechanism for the bump-rebound motion. Tanik and Parlaktas [9] proposed a kinematic model of the double-wishbone suspension for the kinematic analysis. Lee and Shim [10] determined and compared the roll centers

of three planar half-car models using the Aronhold-Kennedy theorem. Kim et al. [11] suggested a new Jacobian approach to the kinestatic analysis of a planar half-vehicle model. Meanwhile, a few studies have developed the dynamic analysis methods for a suspension mechanism. Balike et al. [12] proposed a kinematic-dynamic quarter-vehicle model of a double-wishbone suspension in the dynamic analysis. Hurel et al. [13] introduced a new method to the dynamic analysis of a MacPherson suspension, in which the suspension kinematics was taken into account. Although the suspension mechanism was modelled in these vehicle models, the kinestatics of the suspension mechanism was linearized in the dynamic analysis. Clearly, the effectiveness of the vehicle dynamic models is reduced for the automated driving.

In this paper, it presents a new method to the dynamic analysis of a planar quarter-vehicle model based on the nonlinear kinestatics of the suspension mechanism. The method is unique in a sense that it is unified approach to describe the 1-DOF planar suspension as instantaneous screw. First, the Jacobian is introduced to the kinestatic analysis of the planar serial and parallel mechanisms. Then, the kinematic and static relations of a planar suspension are described using the instantaneous screws. Considering these nonlinear kinestatic relations, the position analysis, velocity analysis and forces analysis of a quarter-vehicle model are carried out. Thereby, the dynamic equations of the quarter-vehicle model are formulated by Lagrangian function. Finally, the effectiveness of the proposed method is confirmed through the comparison analysis with the simulation software Adams/View under different road disturbances.

## 2. Kinestatic Relations of a Planar Mechanism

In plane, a kinematic joint (or pair) can be expressed in Plücker's axis coordinates as $\hat{S} = \begin{bmatrix} r \times s \\ s \end{bmatrix} \in R^{3 \times 1}$, where $s$ is the joint's unit direction vector and $r$ is the joint's position vector, respectively. For prismatic joint (P-joint), it can be represented by a free vector as $\hat{S} = \begin{bmatrix} c & s & 0 \end{bmatrix}^T$, where $c$ and $s$ are the direction cosine and sine of the P-joint. For revolute joint (R-joint), it can be expressed by a unit line vector as $\hat{S}_i = \begin{bmatrix} y_i & -x_i & 1 \end{bmatrix}^T$, where $x_i$ and $y_i$ denote the position coordinates of R-joint. Meanwhile, a general force can be represented by a wrench $\hat{w}$ and written in the Plücker's ray coordinates as $\hat{w} = f\hat{s} = f \begin{bmatrix} s \\ r \times s \end{bmatrix} \in R^{3 \times 1}$, where $f$ is the magnitude, $s$ is the unit direction vector and $r$ is the position vector, respectively. For a pure moment, it can be represented as $\hat{w} = \begin{bmatrix} 0 & 0 & M_o \end{bmatrix}^T$, where $M_o$ is the moment about the origin $O$.

### 2.1. Kinematic Analysis of a Planar Serial Mechanism

Referring to Figure 1, the twist of the last link in $n$-DOF ($1 \leq n \leq 3$) planar serial mechanism can be expressed by

$$\hat{T} = \mathbf{J}_s \dot{q}, \tag{1}$$

where $\dot{q} = \begin{bmatrix} \dot{q}_1 & \cdots & \dot{q}_n \end{bmatrix}^T \in R^{n \times 1}$ is the velocity vector and $\mathbf{J}_s = \begin{bmatrix} \hat{S}_1 & \cdots & \hat{S}_n \end{bmatrix}$ is the screw-based Jacobian. Meanwhile, the twist $\hat{T}$ can also be viewed as $\hat{T} = \lim\limits_{\delta t \to 0} \frac{\delta \hat{D}}{\delta t}$ for the infinitesimal time interval $\delta t$. From Equation (1) the infinitesimal displacement $\delta \hat{D}$ can be written as

$$\delta \hat{D} = \mathbf{J}_s \delta q, \tag{2}$$

where $\delta q = \begin{bmatrix} \delta q_1 & \cdots & \delta q_n \end{bmatrix}^T$. The reciprocal Jacobian is defined as $\mathbf{J}_{rs} = \begin{bmatrix} \hat{r}_1 & \cdots & \hat{r}_n \end{bmatrix}$, where $\hat{r}_i$ is reciprocal to the columns $\hat{S}_j$ of $\mathbf{J}_s$ except the ith column $\hat{S}_i$, i.e., $\hat{r}_i^T \hat{S}_j = 0$ ($i \neq j$). The unit line vector $\hat{r}_i$ passes through the two joints except the ith joint. The inverse kinematic relations of the serial mechanism can be found as

$$\dot{q} = \text{diag}\left( \frac{1}{\hat{r}_1^T \hat{S}_1} \quad \cdots \quad \frac{1}{\hat{r}_n^T \hat{S}_n} \right) \mathbf{J}_{rs}^T \hat{T}, \tag{3}$$

and

$$\delta q = \operatorname{diag}\left(\frac{1}{\hat{r}_1^T \hat{s}_1} \quad \cdots \quad \frac{1}{\hat{r}_n^T \hat{s}_n}\right) \mathbf{J}_{rs}^T \delta \hat{D}. \tag{4}$$

It is assumed that a wrench $\hat{w}$ acts on the last link, as shown in Figure 1. The joint forces (or torques) $\tau$ can be found as

$$\tau = \mathbf{J}_s^T \hat{w}, \tag{5}$$

where $\tau \equiv \begin{bmatrix} \tau_1 & \cdots & \tau_n \end{bmatrix}^T \in R^{n \times 1}$. From Equation (5), the forward static relation of the planar serial mechanism can be written as

$$\hat{w} = \mathbf{J}_{rs} \operatorname{diag}\left(\frac{1}{\hat{r}_1^T \hat{s}_1} \quad \cdots \quad \frac{1}{\hat{r}_n^T \hat{s}_n}\right) \tau. \tag{6}$$

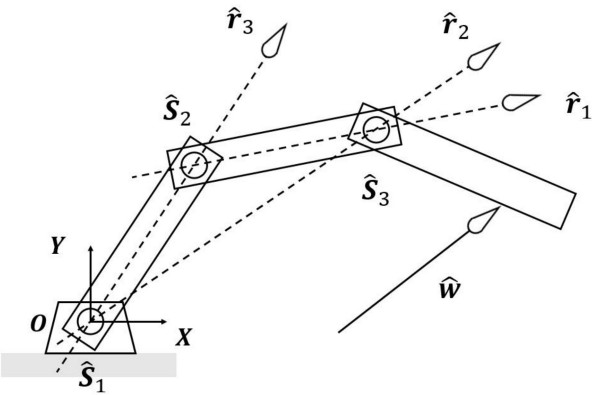

**Figure 1.** A planar serial mechanism.

### 2.2. Kinestatic Analysis of a Planar Parallel Mechanism

Referring to Figure 2, it is assumed that the P-joints of $n$-DOF parallel mechanism are driven by the actuator. The velocity vector $\dot{q}$ of the P-joints can be found as

$$\dot{q} = \mathbf{J}_p \hat{T}, \tag{7}$$

where $\mathbf{J}_p = \begin{bmatrix} \hat{s}_1 & \cdots & \hat{s}_n \end{bmatrix}^T \in R^{n \times 3}$ and $\hat{T}$ is the twist of the moving platform. The $i$th row vector $\hat{s}_i$ of $\mathbf{J}_p$ is the unit line vector along the $i$th P-joint. The reciprocal Jacobian of the parallel mechanism is given as $\mathbf{J}_{rp} = \begin{bmatrix} \hat{R}_1 & \cdots & \hat{R}_n \end{bmatrix} \in R^{3 \times n}$. The $i$th column vector $\hat{R}_i$ of $\mathbf{J}_{rp}$ is determined as the unit line vector which is reciprocal to the $n-1$ row vectors of $\mathbf{J}_x$ except $\hat{s}_i$, i.e., $\hat{R}_i^T \hat{s}_j = 0$ ($i \neq j$). The forward kinematic relations of the parallel mechanism can be obtained as

$$\hat{T} = \mathbf{J}_{rp} \operatorname{diag}\left(\frac{1}{\hat{s}_1^T \hat{R}_1} \quad \cdots \quad \frac{1}{\hat{s}_n^T \hat{R}_n}\right) \dot{q}, \tag{8}$$

and

$$\delta \hat{D} = \mathbf{J}_{rp} \operatorname{diag}\left(\frac{1}{\hat{s}_1^T \hat{R}_1} \quad \cdots \quad \frac{1}{\hat{s}_n^T \hat{R}_n}\right) \delta q. \tag{9}$$

When the forces (or torques) $\tau$ act along the P-joints, the resultant wrench $\hat{w}$ acting on the moving platform can be found as

$$\hat{w} = \mathbf{J}_p^T \tau. \tag{10}$$

Then, the inverse static relation can be given by reciprocal Jacobian $\mathbf{J}_{rp}$ as

$$\tau = \operatorname{diag}\left(\frac{1}{\hat{s}_1^T \hat{R}_1} \quad \cdots \quad \frac{1}{\hat{s}_n^T \hat{R}_n}\right) \mathbf{J}_{rp}^T \hat{w}. \tag{11}$$

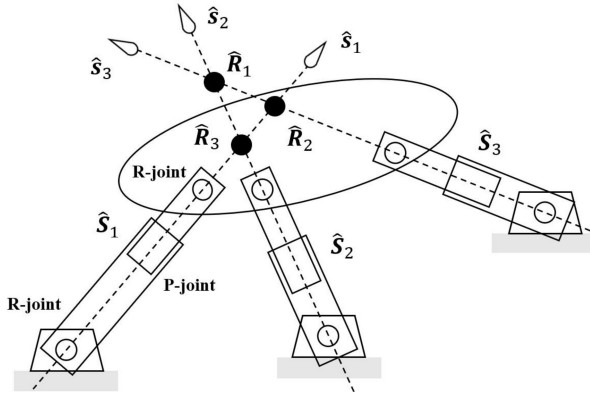

**Figure 2.** A planar parallel mechanism.

### 3. Kinestatic Analysis of a Planar Suspension Mechanism

Referring to Figure 3a, it is assumed that the vehicle body is grounded to describe the relative motions between the wheel and the vehicle body. All the joints $N_i$ ($i = 1$, 2, ..., 7) except $N_2$ are R-joint. The shock absorber is aligned with the P-joint $N_2$. The wheel can be viewed as the moving platform to be connected with the vehicle body by two RR-serial ($N_4N_5$ and $N_6N_7$) and one RPR-serial ($N_1N_2N_3$) kinematic chains. For the RR-chain, there is one constraint wrench $\hat{s}_i$ that is reciprocal to the unit line vectors $\hat{S}_j$ of the R-joints, i.e., $\hat{S}_j^T \hat{s}_i = 0$. The unit line vector $\hat{s}_i$ passes through two R-joints, as shown in Figure 3b. It is also noted that the unit line vectors $\hat{s}_{45}$ and $\hat{s}_{67}$ span a two-dimensional space of constraint wrenches. For the RPR-chain, there is no constraint wrench acting on the wheel and exists a non-constraint wrench $\hat{r}_2$ acting on the wheel. The unit line vector $\hat{r}_2$ passes through the joints $N_1$ and $N_3$ (see in Figure 3b). Then, the wrench acting on the double-wishbone suspension mechanism can be determined from Equation (10) as

$$\hat{w} = f\hat{r}_2, \tag{12}$$

where $f$ is the magnitude of the shock absorber's force. The Jacobian $\mathbf{J}_p$ of the planar suspension mechanism can be expressed as

$$\mathbf{J}_p = [\hat{r}_2]^T \in R^{1 \times 3}.$$

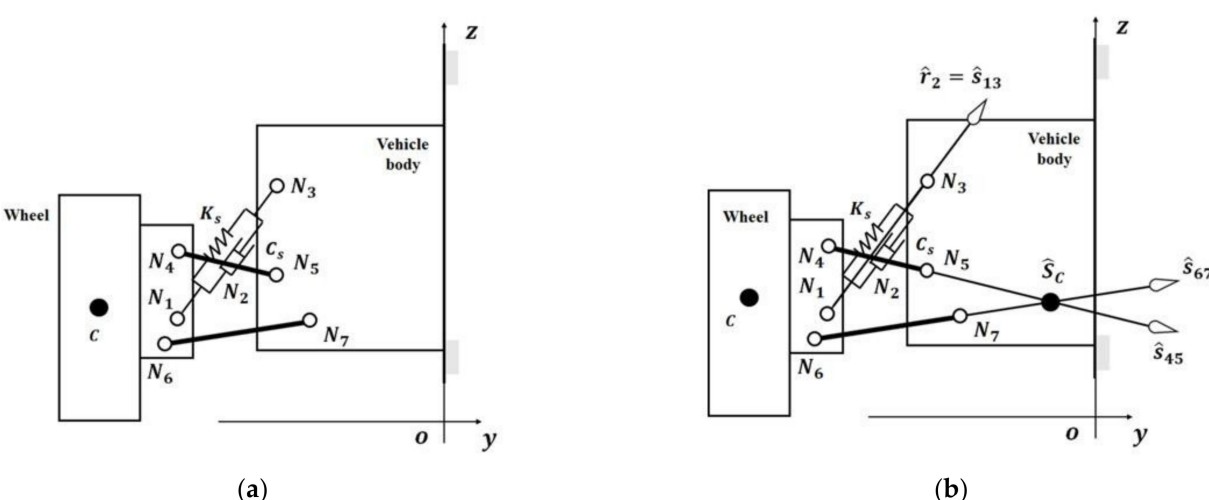

(**a**)                    (**b**)

**Figure 3.** Planar double-wishbone suspension mechanism: (**a**) the schematic diagram; (**b**) the kinematic model.

Due to the two-dimensional space of constraint wrenches, the reciprocal Jacobian $\mathbf{J}_{rp}$ has only one unit column vector $\hat{S}_C$ that is reciprocal to the unit line vectors $\hat{s}_{45}$ and $\hat{s}_{67}$. The inverse static relation can be given by Equation (11)

$$f = \frac{1}{\hat{r}_2^T \hat{S}_C} \hat{S}_C^T \hat{w}. \tag{13}$$

The wheel can be considered to be connected to the vehicle body by a virtual R-joint $\hat{S}_C$ located on the intersection of the unit line vectors $\hat{s}_{45}$ and $\hat{s}_{67}$. Then, the instantaneous twist of the wheel with respect to the vehicle body can be expressed by Equation (8)

$$\hat{T} = \omega \hat{S}_C, \tag{14}$$

where $\omega = \frac{v_d}{\hat{r}_2^T \hat{S}_C}$ represents the angular velocity of the virtual R-joint and $v_d$ is the velocity of the P-joint $N_2$. In addition, since the planar MacPherson suspension mechanism is 1-DOF parallel mechanism, the static and kinematic relations can also be described using Jacobian.

## 4. Dynamic Analysis of a Planar Suspension Mechanism

Referring to Figure 4, a quarter-vehicle model, which consists of the wheel, double-wishbone suspension and vehicle body, is modelled for the dynamic analysis of a vehicle. It is assumed that the tire is connected to the ground by a combination of a vertical spring and damper. When a vehicle runs on the uneven road, the road disturbance is inputted into the wheel. In order to describe the vertical motion of the vehicle body, the vehicle body is considered to connect to the ground by the prismatic joint $N_8$ along $z$-axis [12–14].

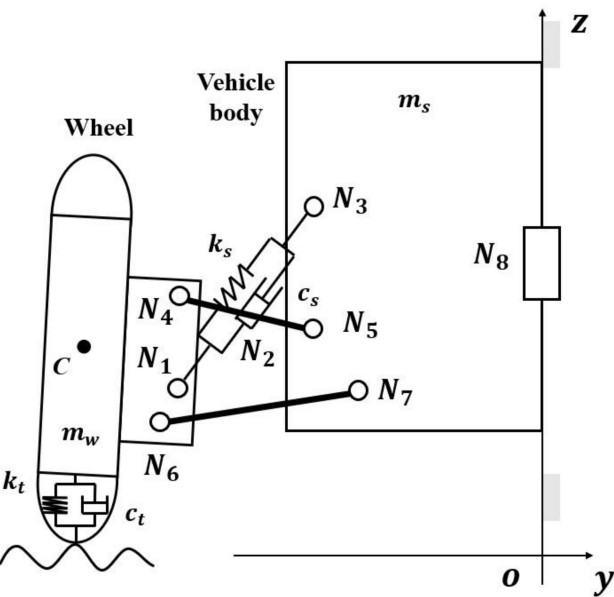

**Figure 4.** Quarter-vehicle model with a double-wishbone suspension.

### 4.1. Position Analysis of a Planar Suspension Mechanism

In Figure 3a, it is assumed that the vehicle body is fixed on the ground. For a finite displacement of the wheel, the general displacement matrix $[D]$ can be formulated as

$$[D] = \begin{bmatrix} cos\theta & -sin\theta & y \\ sin\theta & cos\theta & z \\ 0 & 0 & 1 \end{bmatrix},$$

where $\theta$, $y$ and $z$ are the rotation and the translation of the wheel, respectively. Then, the positions of $N_1$, $N_4$, $N_6$ and the wheel center $C$ can be derived as

$$
\begin{bmatrix}
y_{N1} & y_{N4} & y_{N6} & y_C \\
z_{N1} & z_{N4} & z_{N6} & z_C \\
1 & 1 & 1 & 1
\end{bmatrix}
= [\boldsymbol{D}]
\begin{bmatrix}
y'_{N1} & y'_{N4} & y'_{N6} & y'_c \\
z'_{N1} & z'_{N4} & z'_{N6} & z'_c \\
1 & 1 & 1 & 1
\end{bmatrix},
\tag{15}
$$

where the coordinates $(y'_{N1}, z'_{N1})$, $(y'_{N4}, z'_{N4})$, $(y'_{N6}, z'_{N6})$ and $(y'_C, z'_C)$ are the initial positions of the joints $N_1$, $N_4$, $N_6$ and the wheel center $C$. For the constant lengths of the links $N_4N_5$ and $N_6N_7$, it satisfies that

$$
l_{N4N5} = \left[ \left( y'_{N4} - y_{N5} \right)^2 + \left( z'_{N4} - z_{N5} \right)^2 \right]^{\frac{1}{2}},
$$

and

$$
l_{N6N7} = \left[ \left( y'_{N6} - y_{N7} \right)^2 + \left( z'_{N6} - z_{N7} \right)^2 \right]^{\frac{1}{2}}.
$$

If the vertical displacement $z'_w$ of the wheel with respect to the vehicle body is known, the constraint equations can be formulated as

$$
z_C - z'_C = z'_w,
\tag{16}
$$

$$
\left[ \left( y_{N4} - y_{N5} \right)^2 + \left( z_{N4} - z_{N5} \right)^2 \right]^{\frac{1}{2}} = l_{N4N5},
\tag{17}
$$

$$
\left[ \left( y_{N6} - y_{N7} \right)^2 + \left( y_{N6} - y_{N7} \right)^2 \right]^{\frac{1}{2}} = l_{N6N7}.
\tag{18}
$$

Substituting Equation (15) into the nonlinear Equations (16)–(18), the constraint equations can be written in terms of three unknown parameters $\begin{bmatrix} y & z & \theta \end{bmatrix}$. Using Newton–Raphson method, the displacement matrix $[\boldsymbol{D}]$ can be calculated from the above constraint equations. Then, the planar suspension mechanism can be determined by Equation (15). Meanwhile, the change $\delta l_s$ in the strut spring's length can be obtained as

$$
\delta l_s = l_{N1N3} - l'_{N1N3},
$$

where $l_{N1N3}$ and $l'_{N1N3}$ are the length and initial length of the strut spring, respectively.

### 4.2. Velocity Analysis of a Planar Suspension Mechanism

Referring to Figure 5, the vertical motion of the vehicle body can be expressed as

$$
\hat{\boldsymbol{T}}_s = \dot{z}_s \begin{bmatrix} 0 & 1 & 0 \end{bmatrix}^T,
\tag{19}
$$

where $\dot{z}_s$ is the vertical velocity of the vehicle body. Since it is assumed that the wheel is connected to the vehicle body by the virtual R-joint $\hat{S}_C$, the instantaneous twist of the wheel can be found as

$$
\hat{\boldsymbol{T}}_w = \hat{\boldsymbol{T}}_s - \dot{q}_c \, \hat{S}_C,
\tag{20}
$$

where $\dot{q}_c$ is the angular velocity of the virtual R-joint. Then, the wheel's velocity can be given as

$$
\dot{y}_w = \hat{s}_{wy}^T \hat{\boldsymbol{T}}_w,
\tag{21}
$$

$$
\dot{z}_w = \hat{s}_{wz}^T \hat{\boldsymbol{T}}_w,
\tag{22}
$$

where $\hat{s}_{wy}$ and $\hat{s}_{wz}$ denote the unit line vectors passing through the wheel center $C$ along the $y$ and $z$ axes, respectively. By substituting Equations (19) and (20) into Equation (22), the angular velocity $\dot{q}_c$ of the virtual R-joint can be obtained as

$$
\dot{q}_c = \frac{1}{\hat{s}_{wz}^T \hat{S}_C} \left( \dot{z}_s - \dot{z}_w \right).
\tag{23}
$$

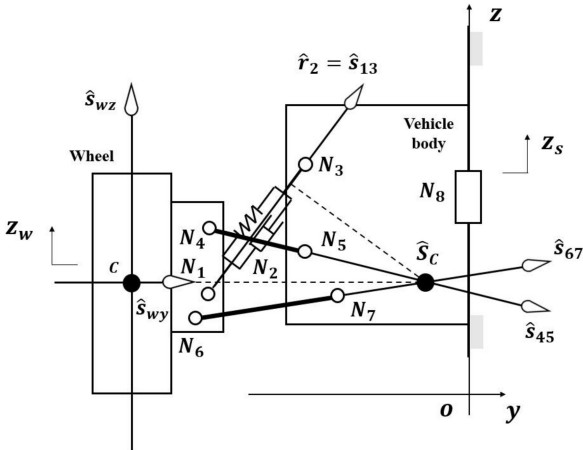

**Figure 5.** Instantaneous screws acting on a quarter-vehicle model.

Then, the velocity $\dot{y}_w$ can be given by Equations (21) and (23)

$$\dot{y}_w = -\frac{\hat{s}_{wy}^T \hat{S}_C}{\hat{s}_{wz}^T \hat{S}_C}(\dot{z}_s - \dot{z}_w). \tag{24}$$

From Equation (20) the angular velocity $\dot{\theta}_w$ of the wheel with respective to the vehicle body can be expressed as

$$\dot{\theta}_w = -\dot{q}_c = -\frac{1}{\hat{s}_{wz}^T \hat{S}_C}(\dot{z}_s - \dot{z}_w). \tag{25}$$

*4.3. Force Analysis of a Planar Suspension Mechanism*

When a vehicle moves on the uneven road, the road disturbance $z_o$ is inputted into the wheel, as shown in Figure 6. For the equivalent stiffness $k_t$ and damping rate $c_t$ of the tire, the vertical force of the tire can be found as

$$\hat{w}_t = \left[-k_t(z_w - z_o) - c_t(\dot{z}_w - \dot{z}_o)\right] \hat{r}_w, \tag{26}$$

where $z_w$ is the vertical displacement of the wheel. In Figure 6, the shock absorber is set between the wheel and vehicle body along P-joint $N_2$. The forces of a strut spring and damper acting on the wheel is given as

$$\hat{w}_2 = (f_s + f_d)\hat{r}_2^T, \tag{27}$$

where $f_s = k_s \delta l_s$, $f_d = c_s v_d$, $v_d = \dot{q}_c \hat{r}_2^T \hat{S}_C$. Since the wheel is assumed to connect to the vehicle body by a virtual R-joint $\hat{S}_C$, the input torque $\tau_C$ about the virtual R-joint is obtained as

$$\tau_C = \hat{w}_2^T \hat{S}_C = (f_s + f_d)\hat{r}_2^T \hat{S}_C. \tag{28}$$

Then, the vertical force acting on the mass center $C$ of the wheel can be given by Equation (28)

$$\hat{w}_C = \frac{\tau_C}{\hat{r}_w^T \hat{S}_C}\hat{r}_w = (f_s + f_d)\frac{\hat{r}_2^T \hat{S}_C}{\hat{r}_w^T \hat{S}_C}\hat{r}_w. \tag{29}$$

Then, the resultant wrench $\hat{w}_w$ acting on the wheel can be given by Equations (26) and (29)

$$\hat{w}_w = \hat{w}_t + \hat{w}_C = F_w \hat{r}_w, \tag{30}$$

where $F_w = -k_t(z_w - z_o) - c_t(\dot{z}_w - \dot{z}_o) + (f_s + f_d)\frac{\hat{r}_2^T \hat{s}_C}{\hat{r}_w^T \hat{s}_C}$. Meanwhile, the vertical force acting on the vehicle body can be expressed as

$$\hat{w}_b = -\hat{w}_C = F_s \hat{r}_w, \tag{31}$$

where $F_s = -(f_s + f_d)\frac{\hat{r}_2^T \hat{s}_C}{\hat{r}_w^T \hat{s}_C}$.

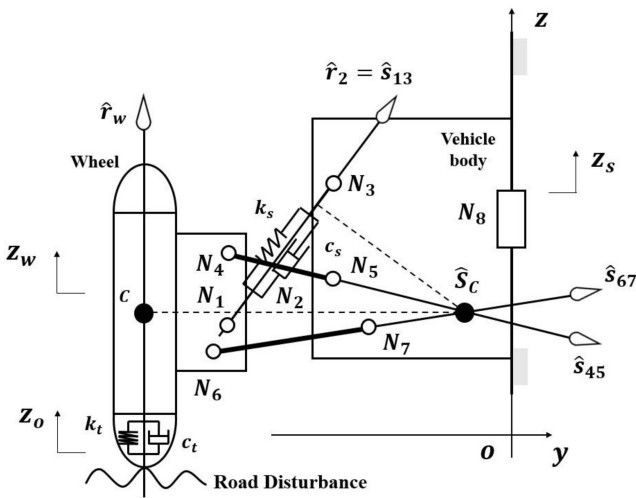

**Figure 6.** Wrenches acting on a quarter-vehicle model.

### 4.4. Dynamic Analysis of a Planar Quarter-Vehicle Model

In this section, Lagrangian equations are employed to describe the dynamic performances of a quarter-vehicle model. As shown in Figure 7, the kinetic energy of the wheel and the vehicle body, respectively, can be found as

$$K_w = \frac{1}{2}m_w\left(\dot{y}_w^2 + \dot{z}_w^2\right) + \frac{1}{2}I_{wx}\dot{\theta}_w^2, \tag{32}$$

$$K_s = \frac{1}{2}m_s\dot{z}_s^2, \tag{33}$$

where $m_s$ and $m_w$ are the masses of the vehicle body and wheel and $I_{wx}$ is the moment of inertia of the wheel. Lagrangian function is given by Equations (24), (25), (32) and (33)

$$L = \frac{1}{2}m_s\dot{z}_s^2 + \frac{1}{2}m_w\dot{z}_w^2 + \frac{1}{2}\left(m_w a_1^2 + I_{wx}a_2^2\right)(\dot{z}_s - \dot{z}_w)^2, \tag{34}$$

where $a_1 = -\frac{\hat{s}_{wy}^T \hat{s}_C}{\hat{s}_{wz}^T \hat{s}_C}$ and $a_2 = -\frac{1}{\hat{s}_{wz}^T \hat{s}_C}$. From Equation (34), the vector of generalized coordinates and the vector of generalized forces can be defined as $q = \begin{bmatrix} z_s & z_w \end{bmatrix}^T$ and $Q = \begin{bmatrix} F_s & F_w \end{bmatrix}^T$, respectively. Then, the system of dynamic equations can be derived from Equations (30), (31) and (34)

$$\begin{bmatrix} b_{11} & b_{12} \\ b_{21} & b_{22} \end{bmatrix}\begin{bmatrix} \ddot{z}_s \\ \ddot{z}_w \end{bmatrix} = Q, \tag{35}$$

where $b_{11} = b_{22} = m_s + (m_w a_1^2 + I_{wx}a_2^2)$, $b_{12} = b_{21} = -(m_w a_1^2 + I_{wx}a_2^2)$, $F_s = -(f_s + f_d)\frac{\hat{r}_2^T \hat{s}_C}{\hat{r}_w^T \hat{s}_C}$ and $F_w = -k_t(z_w - z_o) - c_t(\dot{z}_w - \dot{z}_o) + (f_s + f_d)\frac{\hat{r}_2^T \hat{s}_C}{\hat{r}_w^T \hat{s}_C}$. Then, the dynamic characteristics of a quarter-vehicle model can be described by Equation (35).

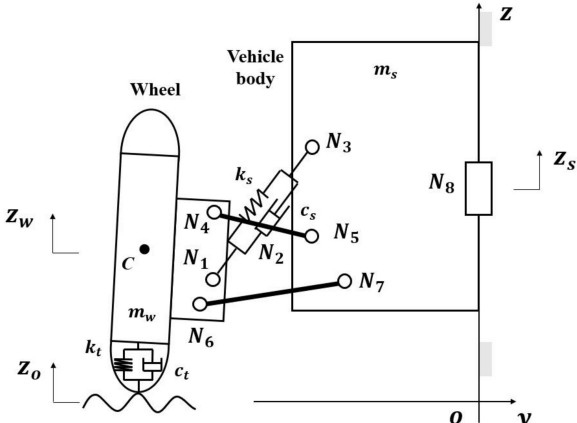

**Figure 7.** Planar quarter-vehicle dynamic model.

## 5. Numerical Example

Referring to Figure 7, a planar double-wishbone suspension mechanism is employed to connect the wheel to the vehicle body in the quarter-vehicle dynamic model. The physical parameters of a quarter-vehicle model are listed in Tables 1 and 2.

**Table 1.** Physical parameters of a quarter-vehicle model.

| Parameters | Value |
|---|---|
| Sprung mass, $m_s$ (kg) | 439.38 |
| Wheel mass, $m_w$ (kg) | 42.27 |
| Suspension spring stiffness, $k_s$ (N/m) | 38,404 |
| Suspension damping rate, $c_s$ (Ns/m) | 3593.4 |
| Tire vertical stiffness, $k_t$ (kN/m) | 200 |
| Tire vertical damping rate, $c_t$ (Ns/m) | 352.27 |

**Table 2.** Initial position of a planar suspension mechanism.

| Joint | Positions (mm) | |
|---|---|---|
| | y | z |
| $N_1$ | −750 | 300 |
| $N_2$ | Prismatic joint | |
| $N_3$ | −450 | 900 |
| $N_4$ | −720 | 600 |
| $N_5$ | −450 | 510 |
| $N_6$ | −840 | 150 |
| $N_7$ | −300 | 240 |
| $N_8$ | Prismatic joint | |
| Wheel center $C$ | −960 | 350 |

### 5.1. Analysis Procedure of a Quarter-Vehicle Model

In this section, it describes the kinestatic relations of the quarter-vehicle model in the initial state. Referring to Figure 3b, the constraint wrenches $\hat{s}_{45}$ and $\hat{s}_{67}$ can be given as

$$\hat{s}_{45} = \begin{bmatrix} 0.95 & -0.32 & -341.53 \end{bmatrix}^T,$$

and

$$\hat{s}_{67} = \begin{bmatrix} 0.99 & 0.16 & -286.05 \end{bmatrix}^T.$$

Since the unit line vector $\hat{S}_C$ is reciprocal to the constraint wrenches $\hat{s}_{45}$ and $\hat{s}_{67}$, it satisfies that $\hat{s}_{45}^T \hat{S}_C = 0$ and $\hat{s}_{67}^T \hat{S}_C = 0$. Then, the unit line vector $\hat{S}_C$ can be determined as

$$\hat{S}_C = \begin{bmatrix} 313.33 & -140 & 1 \end{bmatrix}^T.$$

The unit line vector $\hat{r}_2$ passes through the R-joints $N_1$ and $N_3$ and can be written as

$$\hat{r}_2 = \begin{bmatrix} 0.445 & 0.89 & -804.98 \end{bmatrix}^T.$$

In Figure 5, the unit line vectors $\hat{s}_{wy}$ and $\hat{s}_{wz}$ pass through the wheel center $C$ along the $y$ and $z$ axes, respectively. We can find as

$$\hat{s}_{wy} = \begin{bmatrix} 1 & 0 & -350 \end{bmatrix}^T,$$

and

$$\hat{s}_{wz} = \begin{bmatrix} 0 & 1 & -960 \end{bmatrix}^T.$$

From Equations (24) and (25), the velocity $\dot{y}_w$ and angular velocity $\dot{\theta}_w$ of the wheel are given as

$$\dot{y}_w = -0.03(\dot{z}_s - \dot{z}_w),$$

and

$$\dot{\theta}_w = 9.09 \times 10^{-4}(\dot{z}_s - \dot{z}_w).$$

As shown in Figure 6, the tire's vertical force acts along the unit line vector $\hat{r}_w$, which can be expressed as

$$\hat{r}_w = \begin{bmatrix} 0 & 1 & -960 \end{bmatrix}^T.$$

The magnitudes of the vertical forces acting on the wheel and vehicle body can be given by Equations (30) and (31)

$$F_w = -200(z_w - z_o) - 0.35(\dot{z}_w - \dot{z}_o) + 0.72(f_s + f_d),$$

and

$$F_s = -0.72(f_s + f_d).$$

Substituting $\dot{y}_w$, $\dot{\theta}_w$, $F_w$ and $F_s$ into Equation (35), the dynamic performances of a quarter-vehicle model can be simulated. It is also noted that the dynamic characteristics of a planar MacPherson suspension mechanism can also be described in the same manner.

*5.2. Comparison Analysis of a Quter-Vehicle Model*

In this section, the dynamic responses of a quarter-vehicle model are simulated by the simulation software Adams/View, as shown in Figure 8. The simulation results from the proposed method are compared with the simulation software Adams/View.

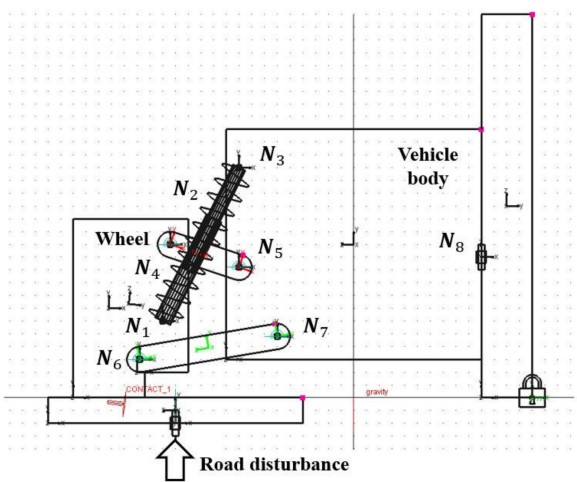

**Figure 8.** Quarter-vehicle model in Adams/View.

### 5.2.1. Harmonic Inputs

Referring to Figure 9, the road disturbance [12] is described using a sinusoidal function

$$z_o = z_{omax}\sin(2\pi ft),$$

where $z_{omax} = 0.05$ m is the amplitude and $f = 1$ Hz is the frequency. In Figure 10, the dynamic responses of the vehicle body are obtained using the proposed method. Compared with the simulation software Adams/View, the vertical displacement ($z_s$) and velocity ($\dot{z}_s$) of the vehicle body (1–5 s) are evaluated using the root mean square error (RMSE) and the integral of squared error (ISE), as shown in Table 3.

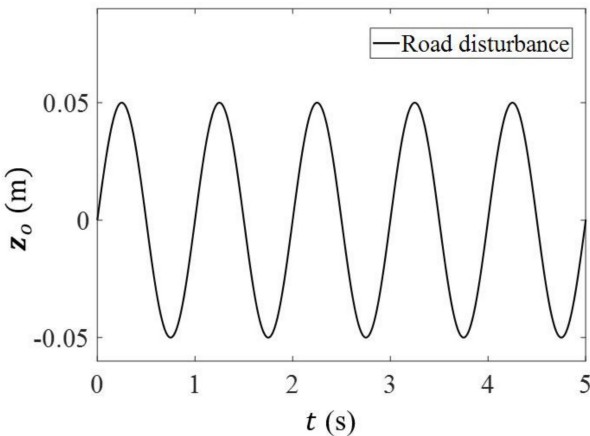

**Figure 9.** Sinusoidal road disturbance: $z_{omax} = 0.05$ m and $f = 1$ Hz.

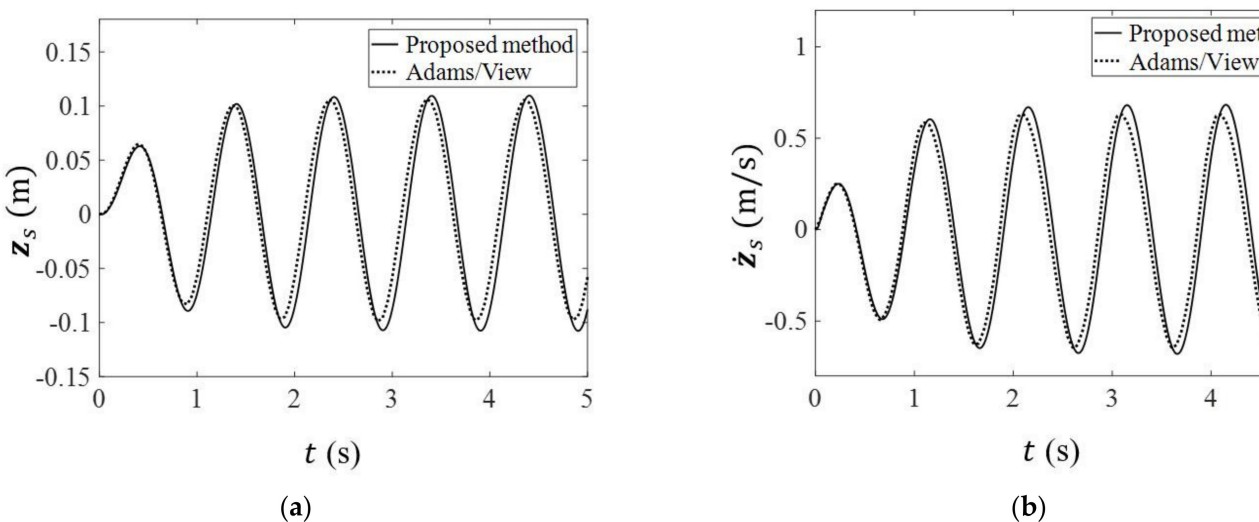

**Figure 10.** Dynamic responses of vehicle body: (**a**) vertical displacement; (**b**) vertical velocity.

**Table 3.** Dynamic responses of the vehicle body compared with the software Adams/View.

| Dynamic Responses of the Vehicle Body | RMSE | ISE |
|:---:|:---:|:---:|
| $z_s$ (m) | 0.0145 | 0.0011 |
| $\dot{z}_s$ (m/s) | 0.0160 | 0.0013 |

### 5.2.2. Single Bump

For a single bump, the road disturbance [15,16] can be described as

$$z_o = \begin{cases} 0.02(1 - cos8\pi t) & 0.5 \le t \le 0.75 \\ 0 & otherwise \end{cases}$$

As shown in Figure 11, a vehicle passes over a single bump with the velocity of 60 km/h. The dynamic responses of a vehicle body are simulated as shown in Figure 12. Compared with the simulation software Adams/View, the characteristic values of the dynamic responses of the vehicle body are listed in Table 4. It is observed that there are only minor differences in the dynamic responses of the vehicle body between the proposed method and the simulation software Adams/View.

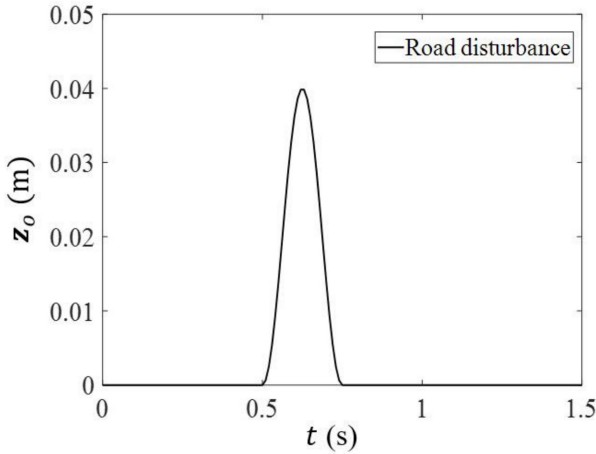

**Figure 11.** Road disturbance of a single bump.

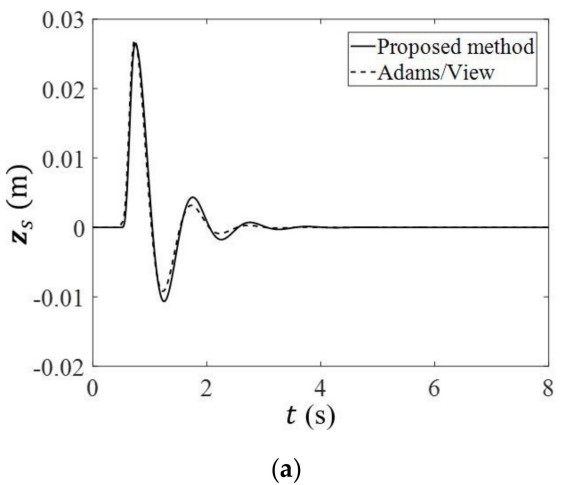

(**a**)

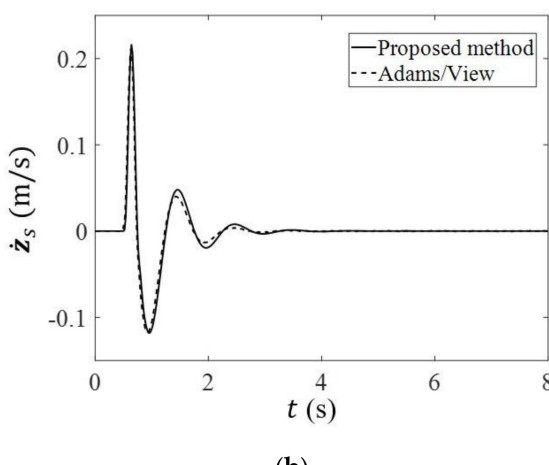

(**b**)

**Figure 12.** Dynamic responses of vehicle body for a single bump: (**a**) ver-tical displacement; (**b**) vertical velocity.

**Table 4.** Characteristic values of the dynamic responses of the vehicle body.

|  | Peak | | | Settling Time (s) | | |
|---|---|---|---|---|---|---|
|  | Adams | Proposed | Error | Adams | Proposed | Error |
| $z_s$ (m) | 0.0270 | 0.0266 | 1.50% | 2.78 | 2.84 | 2.16% |
| $\dot{z}_s$ (m/s) | 0.2112 | 0.2159 | 2.21% | 2.59 | 2.63 | 1.54% |

## 6. Conclusions

This work presented a unified analysis method to describe the dynamic characteristics of a planar suspension mechanism. First, the kinestatic relations of a planar suspension mechanism are described using a Jacobian approach. The suspension can be viewed as a virtual revolute joint to connect between the vehicle body and wheel. Based on these kinestatic relations, it carried out the position analysis, velocity analysis and force analysis for the quarter-vehicle model in sequence. The Lagrangian function is used to formulate the dynamic equations in terms of generalized coordinates. Furthermore, as an example, the dynamic responses of the quarter-vehicle model with a double-wishbone suspension are numerically simulated using the theoretical method and the simulation software Adams/View, respectively. Through the comparison analysis of the simulation results, the validity of the theoretical method is confirmed. In practice, the theoretical method can be used to describe the dynamic characteristics of a quarter-vehicle model more accurately for the design of passive suspension mechanism and the control systems of the semi-active and active suspensions. In additional, the roll motion is also one of the most important motions of a vehicle in cornering. The suspension mechanism affects the roll motion of a vehicle strongly and prevents the vehicle rollover. Since the proposed method is mainly used to describe the vertical motion of a vehicle running on an uneven road, an effective and complete nonlinear full-vehicle model for the dynamic performances of a vehicle in cornering must also be studied.

**Author Contributions:** Conceptualization, G.Z. and Y.W.; methodology, Z.Z.; software, S.J.; writing—original draft preparation, G.Z.; writing—review and editing, Y.W.; funding acquisition, S.C. All authors have read and agreed to the published version of the manuscript.

**Funding:** This work was supported by Startup Foundation for Young Teachers of Shanghai Ocean University, National Natural Science Foundation of China Under Project 52072243.

**Institutional Review Board Statement:** Not applicable.

**Informed Consent Statement:** Not applicable.

**Data Availability Statement:** Not applicable.

**Conflicts of Interest:** The authors declare no conflict of interest. The funders had no role in the design of the study; in the collection, analyses, or interpretation of data; in the writing of the manuscript, or in the decision to publish the results.

## Nomenclature

| | |
|---|---|
| $\hat{w}$ | Wrench |
| $\hat{s}$ | Unit line vector expressed in the Plücker's ray coordinate |
| $\hat{T}$ | Twist |
| $\hat{S}$ | Unit line vector expressed in the Plücker's axis coordinate |
| $\delta\hat{D}$ | Infinitesimal displacement |
| $\mathbf{J}_s$ | Jacobian of a planar serial mechanism |
| $\mathbf{J}_{rs}$ | Reciprocal Jacobian of a planar serial mechanism |
| $\mathbf{J}_p$ | Jacobian of a planar parallel mechanism |
| $\mathbf{J}_{rp}$ | Reciprocal Jacobian of a planar parallel mechanism |
| $N_i$ | Joints of a planar suspension mechanism |
| $\hat{S}_C$ | Unit line vector of virtual revolute joint |
| $\hat{s}_{45}$ | Unit line vector of constraint wrench |
| $\hat{s}_{67}$ | Unit line vector of constraint wrench |
| $\hat{r}_2$ | Unit line vector of non-constraint wrench |
| $\hat{s}_{wy}$ | Unit line vector along $y$ axis |
| $\hat{s}_{wz}$ | Unit line vector along $z$ axis |
| $z_s$ | Vertical displacement of vehicle body |
| $z_w$ | Vertical displacement of wheel |
| $z_o$ | Road disturbance |

| R-joint | Revolute joint |
| P-joint | Prismatic joint |
| RR-serial chain | Revolute–Revolute serial chain |
| RPR-serial chain | Revolute–Prismatic–Revolute serial chain |

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
