# Peer review of "Dynamic Analysis of a Planar Suspension Mechanism Based on Kinestatic Relations"

_electronics, doi:10.3390/electronics11121856_

Round 1
Reviewer 1 Report
Grammar and syntax can be significantly improved, please consider a language review from a native English speaker.
In the introduction, I am missing the scope and motivation behind this research. The paper starts directly with a literature review. Instead of simply presenting what previous researchers did, highlight what knowledge gap you are aiming to address and how this research can be useful for advancing the field.
Moreover, the conclusions present the main findings from the study without addressing potential limitations or important steps for further research.
Reviewer 2 Report
The paper is well organized; the description of the problem is adequate. This work has a good discussion; the topic is adequate directed in a theoretical way.
To improve this paper, from my point of view, the authors may consider the following:
- Review the article format, there are some details to correct. For example, the name of a figure is on a different page than this one.
- To get a better idea of the effectiveness of the presented method, more tests need to be carried out. For example, with different road disturbance functions.
The error integrals could be used to evaluate differences in the dynamic responses of the vehicle body between the proposed method and the Adams/View simulation software (figure 10).
Round 2
Reviewer 2 Report
The new version of the manuscript has considered the comments of the reviewer.
Author Response
Thank you for your comments.